# Sustained Low Incidence of Severe and Fatal COVID-19 Following Widespread Infection Induced Immunity after the Omicron (BA.1) Dominant in Gauteng, South Africa: An Observational Study

**DOI:** 10.3390/v15030597

**Published:** 2023-02-21

**Authors:** Shabir A. Madhi, Gaurav Kwatra, Jonathan E. Myers, Waasila Jassat, Nisha Dhar, Christian K. Mukendi, Lucille Blumberg, Richard Welch, Alane Izu, Portia C. Mutevedzi

**Affiliations:** 1South African Medical Research Council Vaccines and Infectious Diseases Analytics Research Unit, Faculty of Health Sciences, University of the Witwatersrand, Johannesburg 2193, South Africa; 2Infectious Diseases and Oncology Research Institute, Faculty of Health Sciences, University of the Witwatersrand, Johannesburg 2193, South Africa; 3Centre for Environmental and Occupational Health Research, School of Public Health and Family Medicine, University of Cape Town, Cape Town 7701, South Africa; 4National Health Laboratory Services, National Institute for Communicable Diseases, Johannesburg 2193, South Africa; 5Right to Care, Centurion 0046, South Africa

**Keywords:** COVID-19, sero-survey, immunity, infection fatality risk, Africa

## Abstract

We conducted an epidemiologic survey to determine the seroprevalence of SARS-CoV-2 anti-nucleocapsid (anti-N) and anti-spike (anti-S) protein IgG from 1 March to 11 April 2022 after the BA.1-dominant wave had subsided in South Africa and prior to another wave dominated by the BA.4 and BA.5 (BA.4/BA.5) sub-lineages. We also analysed epidemiologic trends in Gauteng Province for cases, hospitalizations, recorded deaths, and excess deaths were evaluated from the inception of the pandemic through 17 November 2022. Despite only 26.7% (1995/7470) of individuals having received a COVID-19 vaccine, the overall seropositivity for SARS-CoV-2 was 90.9% (95% confidence interval (CI), 90.2 to 91.5) at the end of the BA.1 wave, and 64% (95% CI, 61.8 to 65.9) of individuals were infected during the BA.1-dominant wave. The SARS-CoV-2 infection fatality risk was 16.5–22.3 times lower in the BA.1-dominant wave compared with the pre-BA.1 waves for recorded deaths (0.02% vs. 0.33%) and estimated excess mortality (0.03% vs. 0.67%). Although there are ongoing cases of COVID-19 infections, hospitalization and death, there has not been any meaningful resurgence of COVID-19 since the BA.1-dominant wave despite only 37.8% coverage by at least a single dose of COVID-19 vaccine in Gauteng, South Africa.

## 1. Background

By 14 November 2021, over 40% of the global population had been infected at least once by severe acute respiratory syndrome coronavirus-2 (SARS-CoV-2) [1]. In mid-November 2021, the highly transmissible severe acute respiratory syndrome coronavirus-2 (SARS-CoV-2) Omicron BA.1 (B.1.1.529; henceforth referred to as BA.1) variant of concern (VOC) was identified in Southern Africa, heralding a resurgence of COVID-19 globally [2]. The intrinsic transmissibility of BA.1 is estimated to be twice as high as that of the Delta VOC, which has a basic reproduction rate (*Ro*) of 5–6 [3]. The BA.1 VOC is resistant to the neutralizing activity of antibodies induced by first-generation COVID-19 vaccines and infection by the wild-type virus or earlier VOCs [4]. Nevertheless, CD4^+^ and CD8^+^ T-cell immunity against BA.1 induced by prior infection with SARS-CoV-2 ancestry virus and other variants or by COVID-19 vaccines was relatively conserved [5]. Consequently, BA.1 was associated with high numbers of infections, re-infections, and breakthrough COVID-19 in vaccinated individuals due to evasiveness to neutralizing antibody activity induced by earlier variants and COVID-19 vaccines [6]. Nevertheless, the decoupling of BA.1 infections and severe disease transpired, likely due to the relative preservation of underlying cell-mediated immunity [7].

Approximately three months after the BA.1 dominant wave had subsided in South Africa in April 2022, another infection resurgence occurred, dominated by the BA.4 and BA.5 sub-lineages (herewith referred to as BA.4/BA.5), which manifested relative resistance to the neutralizing activity of the antibody induced by BA.1 infection, more so in the unvaccinated [8]. As of mid-November 2022, further sub-lineages of Omicron have evolved, which are even more antibody-evasive compared with BA.1 [9].

Our previous population-based serosurvey in Gauteng Province, South Africa, reported 73% of the population had acquired SARS-CoV-2 immunity prior to the onset of the BA.1 dominant wave despite only 19% having received a single dose of a COVID-19 vaccine [10].

In this study, we report on a follow-up serosurvey conducted after the BA-1 dominant wave had subsided and which coincided with the onset of the BA.4/BA.5 dominant wave. Using paired serum IgG responses from individuals included in the current and preceding survey, population rates of serologically identified SARS-CoV-2 infections during the BA.1 dominant wave were inferred. We updated incidence rates of cases, hospitalizations and deaths, including the BA.4/BA.5 dominant wave and subsequent ongoing cases due to newer sub-lineages of Omicron in Gauteng (South Africa). Serological and epidemiological data allowed the estimation of ratios of inferred infections to reported cases, hospitalizations and fatalities, as well as infection fatality risk (IFR), for the first three COVID-19 waves cumulatively (pre-BA.1) and the BA.1-dominant wave.

## 2. Methods

### 2.1. Study Setting and Data Collection

A third cross-sectional population-based serosurvey in Gauteng, South Africa, was conducted between 1 March 2022 and 11 April 2022; see Appendix A. The details of the study setting and methods of the survey have been previously described [10,11] and can be found in the Appendix A. The COVID-19 vaccines deployed in South Africa were the messenger ribonucleic acid (mRNA) BNT162.b2 and the non-replicating vector Ad26.CoV2.S vaccines.

### 2.2. Serological Analysis

The serology testing for anti-nucleocapsid (anti-N) and anti-spike (anti-S) IgG was carried out on dried blood spot samples obtained from the participants as previously described [10]. Briefly, binding IgG antibodies were measured by bead-based immunoassays on the Luminex platform to quantify serum IgG binding to full-length spike and nucleocapsid and reported in binding antibody units per millilitre (BAU/mL). The anti-N and anti-S assay was calibrated against a research reagent (NIBSC 20/130) and the first WHO International Standard for anti-SARS-CoV-2 (NIBSC 20/136) for anti-SARS-CoV-2 antibody obtained from the National Institute for Biological Standards and Control, UK.

### 2.3. COVID-19 Data Sources

Data sources were as previously described and included daily recorded COVID-19 cases, hospitalisations and deaths to 17 November 2022 from the National Institute for Communicable Diseases (NICD) in South Africa [12], as well as excess deaths (all excess deaths were assumed to be COVID-19 deaths) until 12 November 2022 from the South African Medical Research Council [13]. The mid-2021 Gauteng province population projections from Statistics South Africa (STATS-SA) were used [14].

### 2.4. Statistical Analysis

The anti-N IgG sensitivity for detecting past infection was previously reported as 58.0%; hence, we used anti-N or anti-S IgG positivity to characterise overall seroprevalence [10]. Anti-S IgG positivity in individuals who received a COVID-19 vaccine (either A26.CoV2.S or BNT162.b2 in South Africa) could be due to vaccination or past infection.

The criteria used to determine serological evidence of SARS-CoV-2 infection during the interval between the pre-BA.1 and post-BA.1-dominant wave serosurveys among those individuals with paired samples from each survey are detailed in Appendix A. In individuals who had not received any COVID-19 vaccine in the interval between the two serosurveys and who were seronegative for anti-N or anti-S IgG in the earlier study, seropositivity for either anti-S or anti-N IgG was defined as seroconversion, respectively. For individuals who were anti-N or anti-S IgG positive in the previous survey, seroresponse was defined by a two-fold or greater increase in anti-N IgG or anti-S IgG between the two time points, respectively. The calculation of serological evidence of presumed BA.1 infections was based on either seroconversion or seroresponse between the survey time points. In individuals with paired samples who received a COVID-19 vaccine in the interval between the two serosurveys, a two-fold increase in anti-N IgG or seroconversion for anti-N IgG (i.e., negative to positive) was used as serological evidence of SARS-CoV-2 infection.

The percentage of seropositivity for either anti-N or anti-S in the COVID-19-unvaccinated individuals in the survey prior to the BA.1-dominant wave, multiplied by the STATS-SA population, yielded the inferred number of infections over the course of the first three COVID-19 waves prior to BA.1. The percentage with serological evidence of infection (composite of seroresponse and seroconversion) in the COVID-19 unvaccinated individuals with paired samples multiplied by the STATS-SA population [14] yielded the inferred number of infections during the BA.1 dominant wave. Inferred numbers of SARS-CoV-2 infections were used to calculate ratios of inferred numbers of infections to recorded COVID-19 cases, hospitalizations, and deaths. Inferred numbers of infections at the population level allowed the direct calculation of IFRs, which are the inverse of the inferred infections: recorded (or excess mortality attributable) COVID-19 deaths ratio. Data were analysed using R v4.1.1 (Vienna, Austria) and STATA v16.1 (College Station, TX, USA).

### 2.5. Survey Ethics

The Human Research Ethics Committee at the University of the Witwatersrand granted a waiver for ethics approval of the survey, which was conducted as part of public health surveillance by the Gauteng Department of Health. All participants were, however, required to provide written informed consent, and individuals within a household were free to decline participation.

## 3. Results

### 3.1. Participants

We surveyed 3345 households, including 1052 (31.4%) enrolled in the previous survey. Dried blood spots were obtained from 7510 individuals, including 2420 (32.2%) with paired samples, as shown in Figure 1 and Appendix A. Those with and without paired samples were similar demographically and regarding seroprevalence; see Appendix A. The median dates of the second and third serosurveys were 14 November 2021 and 20 March 2022, respectively. In individuals with paired samples, the median number of days between sample collection was 128 days (IQR: 122–133). Details of COVID-19 dosing are included in the Appendix A.

We show the flow of participants included in survey analyses from identifying the households and approaching individuals for informed consent through to specimen collection, processing, and data analyses. The final analysis included 7510 individuals in 26 sub-districts. Of note, 4590 individuals from the pre-omicron serosurvey could not be sampled during the post-BA.1-dominant wave because 79 individuals (1.1%) out-migrated, and 37 individuals (0.5%) died between the pre- and post-Omicron BA.1-dominant waves. Furthermore, 726 individuals (10.4%) could not be reached because their households were in estates where access was denied, and 185 refused to participate. Finally, 3563 were unavailable for sampling in the current serosurvey.

### 3.2. Seroprevalence

The seropositivity was higher compared with the preceding serosurvey prevalence, as shown in Appendix A. Overall anti-S or anti-N IgG seropositivity was 90.9% (95% confidence intervals (95% CI): 90.2–91.5), ranging across the five Gauteng districts from 86.3% to 93.0%; Table 1. The seropositivity was lower in the <12 year (84.1%) compared with older age groups (>91%); Table 1. 

Only 29.0% (*n* = 1995/6886) of individuals older than 12 years of age who were eligible to receive a COVID-19 vaccine had received at least a single dose. Seropositivity was slightly higher in individuals older than 12 years who had received at least a single dose of the COVID-19 vaccine (96.1%; 95% CI: 95.2–96.9) compared with unvaccinated individuals (89.5%; 95% CI: 88.6–90.3). Higher seropositivity was evident for the vaccinated compared with the unvaccinated across all age groups > 12 years (see Table 2) and across districts and sub-districts (see Appendix A).

### 3.3. Seroconversion, Seroresponse and Seroreversion

Restricting analyses to individuals with paired samples and no COVID-19 vaccination following the pre-BA.1 serosurvey who were anti-N and anti-S IgG seronegative at the previous survey, 74.9% (95% CI: 71.0–78.5; range 61.5% to 80.5% across the districts) demonstrated seroconversion; see Table 1. High rates of seroconversion were also observed across all stratified age groups, ranging from 66.4% (95% CI: 57.9–74.0) in the >50-year age group to 90.9% (95% CI: 76.4–96.9) in the 12–17-year age-group; see Table 1 and Appendix A.

Based on the composite of seroconversion or seroresponse during the BA.1 dominant wave, 63.9% (95% CI; 61.8–65.9) had serological evidence of SARS-CoV-2 infection, varying from 54.1% to 68.5% across districts. The percentage with serological evidence of SARS-CoV-2 infection ranged from 59.0% (95% CI: 54.9–62.9) in the >50-year age group to 77.5% (95% CI: 70.4–83.3) in the 12-to-17-year age group; see Table 1. Serological evidence of SARS-CoV-2 infection in the BA.1-dominant wave was higher in individuals not vaccinated against COVID-19 (67.0%; 95% CI: 64.6–69.3%) compared with those who had been vaccinated (54.8%; 95% CI: 50.5–59.0%) prior to the onset on the BA.1 wave, including when stratified by age groups eligible for vaccination; see Appendix A. Similar trends were seen for individuals vaccinated in the interval period between the two surveys; see Appendix A.

In individuals with paired samples who were seropositive during the pre-BA.1 serosurvey and did not receive any COVID-19 vaccine between the two sampling points, 20% (179/896) and 4.2% (65/153%) had seroreversion for anti-N and anti-S IgG seropositivity, respectively. Furthermore, 3.1% (49/1591) of these individuals who were seropositive for either anti-N or anti-S IgG at the pre-BA.1 serosurvey tested seronegative for both epitopes in the latest survey.

### 3.4. COVID-19 Rates, Hospitalizations and Deaths

By 6 June 2022, the daily case and hospitalization rates in the BA.4/BA.5-dominant wave had already returned to those of the inter-wave period prior to the onset of the BA.4/BA.5- dominant wave (Figure 2). Compared with the BA.1 dominant wave over a 5-month period (23 October 2021 to 21 March 2022), the subsequent eight-month period (22 March 2000 until 17 November 2022) was associated with an even lower percentage of the total cumulative number of recorded COVID-19 events since the start of the pandemic. Whereas the BA.1-dominant wave contributed to 13.8% and 5.8% of the cumulative number of recorded COVID-19 hospitalizations and deaths, respectively, the corresponding percentages for the eight-month period in the Omicron sub-lineages era were 7.2% and 3.0%; Table 3 and Figure 2. The cumulative excess mortality estimates in the post-BA.1 period (*n* = 6753) was, however, higher than during the BA.1-dominant wave (*n* = 2974) and contributed to 10.2% and 4.5% of excess deaths due to natural causes since the start of the pandemic, respectively.

The cumulative incidence rate (per 100,000) of recorded COVID-19 cases declined from 5957 over the first three COVID-19 waves (pre-BA.1) to 1805 in the BA.1-dominant wave and 523 in the Omicron sub-lineages era. The inferred infections to recorded COVID-19 cases ratio increased from 9.1 in the pre-BA.1 period to 36.3 in the BA.1 dominant wave, indicating greater under-ascertainment of infections in the latter wave; see Table 3 and Figure 2.

The cumulative incidence rate (per 100,000) of COVID-19 hospitalizations declined from 822 in the pre-BA.1 period to 143 and 75 in the BA.1-dominant wave and the Omicron sub-lineages era, respectively. Seven percent of all COVID-19 hospitalizations since the start of the pandemic occurred in the eight-month Omicron sub-lineages era, compared with 13.8% during the BA.1-dominant wave over a five-month period. Whereas an estimated 66 infections resulted in one COVID-19 hospitalization during the pre-BA.1 period, there was one COVID-19 hospitalization for every 458 inferred infections during the BA.1 dominant wave; see Table 3.

For COVID-19 cases, the pre-BA.1-period cumulative wave, the Omicron BA.1-dominant wave, and BA4/5 resurgence occurred from 7 March 2020 to 22 October 2021, 23 October 2021 to 21 March 2022, and 22 March 2022 to 6 June 2022, respectively. For COVID-19 hospitalizations, the pre-BA.1 cumulative wave, the Omicron BA.1-dominant wave, and BA4/5 resurgence occurred from 7 March 2020 to 1 November 2021, 2 November 2021 to 23 March 2022, and 24 March 2022 to 6 June 2022, respectively. For COVID-19 recorded deaths, the pre-BA.1 cumulative wave, the Omicron BA.1-dominant wave, and BA4/5 resurgence occurred from 31 March 2020 to 3 November 2021, 4 November 2021 to 14 April 2022, and 14 April 2022 to 6 June 2022, respectively. For COVID-19 attributable excess deaths, the pre-BA.1 cumulative period, the Omicron BA.1-dominant wave, and BA4/5 resurgence occurred from 3 March 2020 to 27 November 2021, 28 November 2021 to 19 March 2022, and 20 March 2022 to 6 June 2022, respectively.

All rates were smoothed using a 7-day moving average except for excess mortality. The shaded area in grey indicates when the serosurvey sampling was undertaken.

The cumulative incidence rate (per 100,000) of recorded COVID-19 deaths declined from 180.6 to 11.6 and 5.90 during the pre-BA.1, BA.1-dominant wave and the Omicron sub-lineage era, respectively. Ninety-one percent of all recorded deaths since the start of the pandemic preceded the onset of the BA.1 dominant wave, 5.8% occurred during the BA.1-dominant wave, and only 3.0% occurred in the Omicron sub-lineages era. Overall, there was one recorded COVID-19 death for every 300 and 5643 inferred infections, with IFR of 0.33% and 0.02% in the pre-BA.1 period and BA.1-dominant wave, respectively; see Table 3.

The cumulative incidence rate (per 100,000) of COVID-19-attributable deaths using the excess mortality estimates was approximately 2.2 times higher compared with the recorded deaths (425 vs. 191.7, respectively). The cumulative incidence rate of excess deaths declined from 362.6 over the pre-BA.1 period to 19.2 during the BA.1-dominant wave and was 43.6 in the Omicron sub-lineage era. Overall, there was one death for every 149 and 3719 inferred infections, corresponding to IFRs of 0.67% and 0.03% in the pre-BA.1 period and BA.1-dominant wave, respectively; see Table 3.

An age-group-stratified analysis of the cumulative incidence rates showed the same downward trend of recorded COVID-19 cases, hospitalizations and deaths from the pre-BA.1 period through the BA.1-dominant wave to the Omicron sub-lineage era, as shown in Figure 3A–D and Appendix A. In adults older than 50 years of age, the BA.1-dominant wave and Omicron sub-lineage era, respectively, contributed to 9.1% and 6.0% of COVID-19 hospitalizations and 5.0% and 2.8% of recorded deaths that occurred since the start of the pandemic. Additionally, comparing the pre-BA.1 period with the BA.1-dominant wave, the ratio of inferred infections for every COVID-19 hospitalisation increased from 11.8 to 213 and from 38 to 1352 for recorded COVID-19 deaths, corresponding to IFR values of 2.63% and 0.07% (Appendix A).

For COVID-19 cases, the pre-Omicron BA.1 cumulative wave, the Omicron BA.1-dominant wave and the BA4/5 resurgence occurred from 7 March 2020 to 22 October 2021, 23 October 2021 to 21 March 2022, and 22 March 2022 to 6 June 2022, respectively. For COVID-19 hospitalizations, the pre-BA.1 cumulative wave, the Omicron BA.1-dominant wave, and the BA4/5 resurgence occurred from 7 March 2020 to 1 November 2021, 2 November 2021 to 23 March 2022, and 24 March 2022 to 6 June 2022, respectively. For COVID-19 recorded deaths, the pre-BA.1 cumulative wave, Omicron BA.1-dominant wave, and the BA4/5 resurgence were from 31 March 2020 to 3 November 2021, 4 November 2021 to 14 April 2022, and 15 April 2022 to 6 June 2022, respectively. All rates are smoothed using a 7-day moving average. The shaded area in grey indicates when the serosurvey sampling was undertaken.

## 4. Discussion

Despite only 26.7% (1995/7470) of individuals in the survey having received at least a single dose of COVID-19 vaccine at the time of the serosurvey after the BA.1-dominant wave, the overall seropositivity for SARS-CoV-2 was 90.9% after the BA.1-dominant wave had subsided in Gauteng; including 89.5% in COVID-19 unvaccinated individuals older than 12 years of age. Using paired serology data, 63.9% of the population was infected with the BA.1 variant during the fourth COVID-19 wave in Gauteng. Serological evidence of infection during the BA.1-dominant wave was higher (67.0%) in individuals who were unvaccinated against COVID-19 compared with those who had only been vaccinated before the onset of the BA.1-dominant wave (54.8%). Against this background of high seropositivity and high rates of BA.1 infections, we observed even further decoupling of SARS-CoV-2 infection and recorded COVID-19 hospitalizations and deaths during the subsequent emergence of Omicron sub-lineages. Although these resulted in a modest resurgence of COVID-19 infections in Gauteng and in South Africa, the burden of the disease was even further attenuated compared with the BA.1-dominant wave. Although there has been circulation of newer sub-lineages of Omicron since June 2002 in South Africa, including 22E/BQ.1.*, XAY and XBB, BA.5 has remained the dominant circulating sub-lineages of Omicron [15]. The Omicron sub-lineage era contributed to 7.2% and 3.0% of all recorded COVID-19 hospitalizations and deaths (over an eight-month period) since the start of the pandemic, compared with 13.8% and 5.8% of these events during the BA.1-dominant wave (over a five-month period).

We were also able to provide an update on the burden of COVID-19 during the BA.1-dominant wave compared with the earlier three COVID-19 waves. The SARS-CoV-2 cumulative IFR declined by a factor of 16.5, one order of magnitude, from 0.33% to 0.02% for recorded deaths and by a factor of 22.3 from 0.67% to 0.03% for excess deaths in the pre-BA.1 waves compared with the BA.1-dominant wave. In comparison, the IFR for seasonal influenza virus pre-COVID-19 pandemic in South Africa is estimated to be 0.05% based conservatively on approximately 35% (*n* = 20.9 million) of the population with serological evidence of infection and excess deaths attributed to influenza of 11,000 per annum [16,17,18]. We may, however, have overestimated the IFR in the pre-BA.1 period, as the cross-sectional seroprevalence used to infer infections over the course of the first three COVID-19 waves would have missed re-infections.

Notably, there has been a 1.93-fold decrease in recorded COVID-19 deaths over the five-month period that constituted the BA.1-dominant wave (*n* = 1802) compared with the subsequent eight-month period when Omicron sub-lineages have dominated (*n* = 913). The excess deaths, however, were higher during the latter period (*n* = 6753) compared with the BA.1-dominant wave (*n* = 2974). The ratio of excess deaths to recorded deaths increased from 1.65-fold (2974/1802) in the BA.1-dominant wave to 7.40 (6753/913) in the Omicron sub-lineage era. It is unlikely there has been differential under-reporting of COVID-19 deaths between these two periods to explain this discrepancy, as there were no changes in access to healthcare, and DATCOV surveillance has been stable during this time. Rather, these data suggest that excess mortality calculations are becoming less reliable in reflecting the true extent of COVID-19 deaths. This could be due to a number of factors, including the re-emergence of other infectious diseases. The Omicron sub-lineage era in Gauteng coincided with an epidemic of respiratory syncytial virus, as well as an influenza virus epidemic, which occurred earlier than anticipated based on pre-COVID-19 seasonal epidemiology [19]. By contrast, seasonal influenza was largely absent during 2020, and there was only a mild influenza season in 2021 in South Africa. Other chronic causes of death aggravated by the socio-economic fallout of the pandemic and related control measures and reduced effectiveness of the healthcare system during the course of the COVID-19 pandemic could also have contributed to variation over time in the consistency of excess mortality estimates as a proxy for COVID-19 attributable deaths. Other limitations of our study are elaborated upon in the Appendix A.

There is uncertainty as to whether the BA.1 variant is intrinsically less virulent than earlier variants [20,21]. The propensity of BA.1 to infect the upper rather than the lower airways could have contributed to the decoupling of infection and COVID-19 [22,23,24]. The BA.2 sub-lineage was, however, associated with high mortality rates in Hong Kong in populations with low levels of immunity. Nevertheless, our study indicates that unless future variants harbour mutations that evade poly-epitopic CD4^+^ and CD8^+^ immunity induced by current vaccines and past infection, or the virus becomes intrinsically more virulent, COVID-19 no longer poses a major threat of a large burden of severe disease and death compared with the period prior to late 2021 before the evolution of extensive population immunity. Notably, the breadth of T-cell immunity against SARS-CoV-2 is expected to be more diverse in settings such as South Africa, where there has been a high rate of SARS-CoV-2 infection [25]. Poly-epitopic T-cell responses following SARS-CoV-2 infections by all variants to date are directed against the spike, nucleocapsid and membrane protein epitopes. These responses contribute to attenuating the progression of infection to severe disease and may also reduce transmission of the virus [7]. Although hybrid immunity with three doses of COVID-19 vaccines protects better against BA.1-symptomatic COVID-19 than infection-only induced immunity, both are similarly effective against severe COVID-19 [19].

Following the waning of the BA.4/BA.5-dominant wave, South Africa ended all restrictions aimed at limiting the transmission of SARS-CoV-2 on 22 June 2022 [26]. Nevertheless, recorded COVID-19 cases, hospitalizations and death have not shown significant increases since then despite the emergence of further sub-lineages of Omicron, which are even more antibody-evasive and have a better propensity than BA.1 to infect the lower airways [27,28,29]. Additionally, the absence of any further resurgence of COVID-19 since the end of the BA.4/BA.5-dominant wave has persisted despite the vaccine coverage in Gauteng for at least a single dose of COVID-19 vaccine by 14 November 2022 only being 37.8% (*n* = 5,905,254) [30].

The experience in South Africa indicates very high rates of infection and re-infection amongst both the unvaccinated and vaccinated decoupled SARS-CoV-2 infections from severe disease in a population with very high seropositivity prevalence, heralding an endemic phase to the pandemic akin to other endemic respiratory viruses that may cause epidemics, such as seasonal influenza. The deployment of COVID-19 vaccines in Africa has lagged behind other regions, with only 32% of the population having received at least a single dose of COVID-19 vaccine as of December 12, 2022 compared with 69% of the overall global population [31]. Nevertheless, our findings indicate that the strategy and target of the current generation of COVID-19 vaccine rollout need to be reconsidered in Africa, where the seropositivity for SARS-CoV-2 was already 86.7% (84.6–88.5%) in December 2021 [32]. Considering that even up to four doses are sub-optimal in sustaining protection against SARS-CoV-2 infection and non-severe COVID-19 [33], the focus in Africa should be targeting high-risk groups who may remain susceptible to developing severe COVID-19 even after past SARS-CoV-2 infection with or without past vaccination.

## Figures and Tables

**Figure 1 viruses-15-00597-f001:**
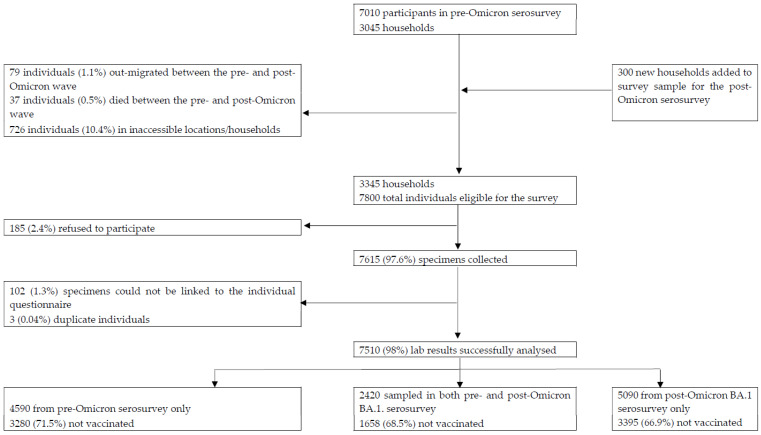
Flow of households and participants included in the seroprevalence surveys.

**Figure 2 viruses-15-00597-f002:**
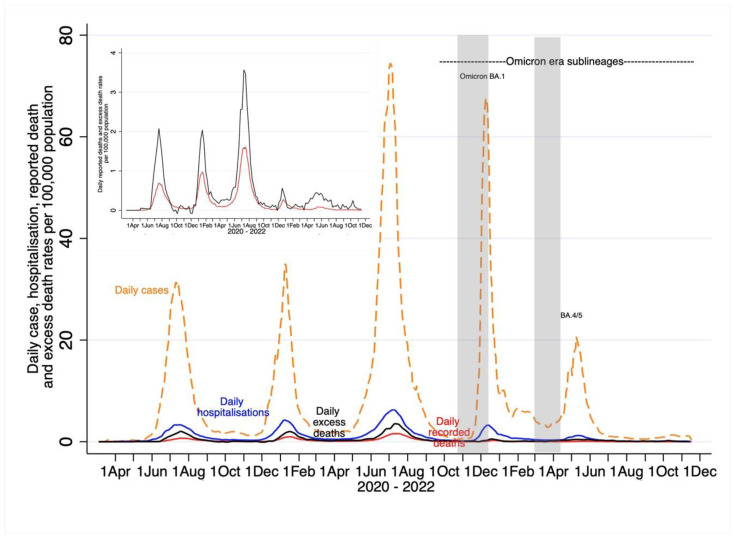
Overall trends of daily incidence per 100,000 recorded COVID-19 cases, hospitalizations and deaths, and excess mortality-attributable COVID-19 deaths for Gauteng, South Africa. Inset figure magnifies COVID-19-recorded deaths and excess mortality.

**Figure 3 viruses-15-00597-f003:**
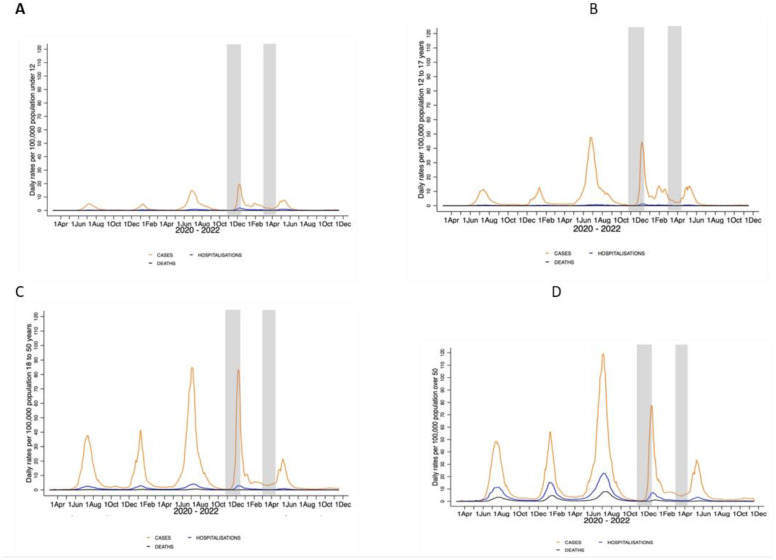
(**A**–**D**): Age-group-stratified analysis of daily moving averages of recorded cases, hospitalizations and deaths in Gauteng Province for each of five COVID-19 waves.

**Table 1 viruses-15-00597-t001:** Seroconversion and changes in SARS-CoV-2 anti-spike (anti-S) or anti-nucleocapsid (anti-N) immunoglobulin G (IgG) Gauteng Province during the Omicron wave in South Africa.

	Pre-BA.1-Dominant Wave Survey		Post-BA.1-Dominant Wave Survey		Individuals with Paired Samples from Pre-BA.1- and Post-BA.1-Dominant Wave Serosurveys and No COVID-19 Vaccination Following Pre-BA.1 Serosurvey ^1^		
District	*N*	Seroprevalence ^1^ *n* (%; 95% CI ^6^)	*N*	Seroprevalence ^2^ *n* (%; 95% CI ^6^)	Seroconversion ^3^ *n*/*N* (%; 95% CI ^6^)	Seroresponse for Anti-N and/or Anti-S IgG ^4^ *n*/*N* (%; 95% CI ^6^)	Overall Serological Evidence SARS-CoV-2 Infection ^5^ *n*/*N* (%; 95% CI ^6^)
Gauteng Province	7010	5124 (73.1; 72.0–74.1)	7510	6823 (90.9; 90.2–91.5)	382/510 (74.9; 71.0–78.5)	933/1548 (60.3; 57.8–62.7)	1315/2058 (63.9; 61.8–65.9)
Johannesburg District	2468	1880 (76.2; 74.5–77.8)	2630	2412 (91.7; 90.6–92.7)	124/154 (80.5; 73.6–86.0)	351/574 (61.1; 57.1–65.1)	475/728 (65.2; 61.7–68.6)
Ekurhuleni District	1861	1382 (74.3; 72.2–76.2)	2132	1982 (93.0; 91.8–94.0)	133/167 (79.6; 72.9–85.0)	344/529 (65; 60.9–69.0)	477/696 (68.5; 65.0–71.9)
Sedibeng District	564	398 (70.6; 66.7–74.2)	624	557 (89.3; 86.6–91.5)	21/30 (70; 52.1–83.3)	44/77 (57.1; 46.0–67.6)	65/107 (60.7; 51.3–69.5)
City of Tshwane District	1464	975 (66.6; 64.1–69.0)	1455	1255 (86.3; 84.4–87.9)	72/117 (61.5; 52.5–69.9)	137/269 (50.9; 45.0–56.8)	209/386 (54.1; 49.2–59.1)
West Rand	653	489 (74.9; 71.4–78.1)	669	617 (92.2; 89.9–94.0)	32/42 (76.2; 61.5–86.5)	57/99 (57.6; 47.7–66.8)	89/141 (63.1; 54.9–70.6)
Age group stratification							
<12 years	753	423 (56.2; 52.6–59.7)	584	491 (84.1; 80.9–86.8)	53/74 (71.6; 60.5–80.6)	82/126 (65.1; 56.4–72.8)	135/200 (67.5; 60.7–73.6)
12 to 17 years	622	459 (73.8; 70.2–77.1)	553	523 (94.6; 92.4–96.2)	30/33 (90.9; 76.4–96.9)	94/127 (74; 65.8–80.9)	124/160 (77.5; 70.4–83.3)
18 to 50 years	4047	2978 (73.6; 72.2–74.9)	4614	4204 (91.1; 90.3–91.9)	210/270 (77.8; 72.4–82.3)	495/836 (59.2; 55.8–62.5)	705/1106 (63.7; 60.9–66.5)
>50 years	1588	1264 (79.6; 77.5–81.5)	1739	1587 (91.3; 89.8–92.5)	85/128 (66.4; 57.9–74.0)	256/450 (56.9; 52.3–61.4)	341/578 (59; 54.9–62.9)

n, number with outcome. N, denominator of population sampled. ^1^ Of the 2420 paired samples available, 2058 did not receive a COVID-19 vaccination between the pre- and post-Omicron BA.1 serosurveys. ^2^ Seroprevalence was defined as seropositive for anti-S or anti-N IgG irrespective of vaccination status. ^3^ Seroconversion is defined for individuals who were seronegative to both S and N on the pre-Omicron serosurvey and seroconverted to either S or N on the post-Omicron BA.1 serosurvey. ^4^ Seroresponse for anti-N and/or anti-S IgG is defined for individuals who were seropositive to either S or N on the pre-Omicron serosurvey and either seroconverted to S, seroconverted to N, or were seropositive to N on the pre-Omicron BA.1 serosurvey and had a ≥2-fold increase in anti-N titers on the post-BA.1-dominant wave serosurvey or were seropositive to S on the pre-BA.1 serosurvey and had a ≥2-fold increase in anti-S titers on the post-BA.1 serosurvey. ^5^ Overall serological evidence of SARS-CoV-2 infection in the period between the two surveys when the BA.1-dominant wave occurred was defined as either seroconversion or seroresponse for anti-N and/or anti-S IgG. ^6^ CI, confidence interval; confidence intervals have not been adjusted for multiplicity and should not be used for inference. The criteria used for determining seroconversion and seroresponse are outlined in Appendix A.

**Table 2 viruses-15-00597-t002:** Seroprevalence of SARS-CoV-2 anti-spike or anti-nucleocapsid immunoglobulin G and risk factors for seropositivity in Gauteng Province stratified by sex, age group, and district.

	Pre-BA.1-Dominant Wave Serosurvey ^1^		Post-BA.1-Dominant Wave Serosurvey	
Category	Number Sampled *N* (%)	Seroprevalence ^2^ *n* (%; 95% CI ^3^) (%; 95% CI)	Number Sampled *N* (%)	Seroprevalence ^2^ *n* (%; 95% CI ^3^)
All participants	7010	5124 (73.1; 72.0–74.1)	7510	6823 (90.9; 90.2–91.5)
Sex				
Male	2941 (42%)	1999 (68; 66.3–69.6)	3096 (41.4%)	2726 (88; 86.9–89.1)
Female	4065 (58%)	3123 (76.8; 75.5–78.1)	4390 (58.6%)	4075 (92.8; 92.0–93.6)
Age group–years				
<12	753 (10.7%)	423 (56.2; 52.6–59.7)	584 (7.8%)	491 (84.1; 80.9–86.8)
12–18	622 (8.9%)	459 (73.8; 70.2–77.1)	553 (7.4%)	523 (94.6; 92.4–96.2)
>18 to 50	4047 (57.7%)	2978 (73.6; 72.2–74.9)	4614 (61.6%)	4204 (91.1; 90.3–91.9)
>50	1588 (22.7%)	1264 (79.6; 77.5–81.5)	1739 (23.2%)	1587 (91.3; 89.8–92.5)
Vaccination status				
Not vaccinated (all age groups)	4938 (70.4%)	3473 (70.3; 69.0–71.6)	4891 (65.5%)	4377 (89.5; 88.6–90.3)
Vaccinated	1319 (18.8%)	1228 (93.1; 91.6–94.3)	1995 (26.7%)	1918 (96.1; 95.2–96.9)
<12 yrs	753 (10.7%)	423 (56.2; 52.6–59.7)	584 (7.8%)	491 (84.1; 80.9–86.8)
Vaccination by age group				
<12 unvaccinated	753 (10.7%)	423 (56.2; 52.6–59.7)	584 (7.8%)	491 (84.1; 80.9–86.8)
12–18 unvaccinated	603 (8.6%)	443 (73.5; 69.8–76.8)	442 (5.9%)	412 (93.2; 90.5–95.2)
12–18 vaccinated	19 (0.3%)	16 (84.2; 62.4–94.5)	106 (1.4%)	106 (100; 96.5–100.0)
>18 to 50 unvaccinated	3356 (47.9%)	2335 (69.6; 68.0–71.1)	3470 (46.5%)	3109 (89.6; 88.5–90.6)
>18 to 50 vaccinated	691 (9.9%)	643 (93.1; 90.9–94.7)	1130 (15.1%)	1082 (95.8; 94.4–96.8)
>50 unvaccinated	979 (14%)	695 (71; 68.1–73.7)	979 (13.1%)	856 (87.4; 85.2–89.4)
>50 vaccinated	609 (8.7%)	569 (93.4; 91.2–95.1)	759 (10.2%)	730 (96.2; 94.6–97.3)
Reported previous COVID-19-positive test				
Never tested	5956 (85%)	4272 (71.7; 70.6–72.9)	7209 (96.4%)	6547 (90.8; 90.1–91.5)
Tested positive	195 (2.8%)	172 (88.2; 82.9–92.0)	43 (0.6%)	43 (100; 91.8–100.0)
Tested negative	859 (12.3%)	680 (79.2; 76.3–81.7)	229 (3.1%)	207 (90.4; 85.9–93.6)
Smoking status				
Non-smoker	4086 (58.3%)	3172 (77.6; 76.3–78.9)	4336 (58%)	3986 (91.9; 91.1–92.7)
Daily	1115 (15.9%)	741 (66.5; 63.6–69.2)	1331 (17.8%)	1173 (88.1; 86.3–89.8)
Once or twice a week	238 (3.4%)	178 (74.8; 68.9–79.9)	393 (5.3%)	361 (91.9; 88.7–94.2)
Occasionally	196 (2.8%)	151 (77; 70.7–82.4)	278 (3.7%)	257 (92.4; 88.7–95.0)
<18 yrs	1375 (19.6%)	882 (64.1; 61.6–66.6)	1137 (15.2%)	1014 (89.2; 87.2–90.9)
Comorbidities				
None	4631 (66.1%)	3432 (74.1; 72.8–75.4)	5233 (70%)	4758 (90.9; 90.1–91.7)
1 or more	1004 (14.3%)	810 (80.7; 78.1–83.0)	1111 (14.9%)	1025 (92.3; 90.5–93.7)
<18 yrs (not assessed)	1375 (19.6%)	882 (64.1; 61.6–66.6)	1137 (15.2%)	1014 (89.2; 87.2–90.9)
HIV status				
HIV-negative	6460 (92.2%)	4728 (73.2; 72.1–74.3)	6850 (91.6%)	6232 (91; 90.3–91.6)
HIV-positive	550 (7.8%)	396 (72; 68.1–75.6)	631 (8.4%)	565 (89.5; 86.9–91.7)

Note: Missing in current serosurvey 3: sex = 24; age group = 20; vaccination status = 40; ever tested COVID = 29, smoke = 35; co-morbidities = 29, and self-reported HIV = 29. Missing in previous serosurvey 2: sex = 4. n, number with outcome. N, denominator of population sampled. ^1^ Madhi et al., 2022 [10]. ^2^ Seroprevalence was defined as seropositive for anti-S or anti-N IgG, irrespective of vaccination status. ^3^ CI, confidence interval; confidence intervals have not been adjusted for multiplicity and should not be used for inference.

**Table 3 viruses-15-00597-t003:** Cumulative reported COVID-19 cases, hospitalizations, recorded deaths, and excess mortality in Gauteng Province by COVID-19 wave.

Outcomes	Pre-BA.1-Dominant Wave Cumulative	BA.1-Dominant Wave	Omicron Sublineage Era	Total
Period of case wave	7 March 2020 to 22 October 2021	23 October 2021 to 21 March 2022	22 March 2022 to 17 November 2022	
Inferred infections from serosurvey ^1^	8,391,304 (8,265,033–8,506,096)	10,167,996 (9,803,769–10,517,047)	Not applicable	
Cases—no. ^†^	926,193	279,829	135,272	1,341,294
Cumulative case rate per 100,000 population	5957	1805	523	8653
Annualised case rate per 100,000 population	3567	1080	313	5182
Proportion of total cumulative cases, %	69.1	20.9	10	100
Inferred infection: recorded case ratio (95% CI)	9.1 (8.9–9.2).	36.3 (35.0–37.6)	Not applicable	
Period of COVID-19 hospitalisation wave	7 March 2020 to November 1, 2021	2 November 2021 to 23 March 2022	24 March 2022, 2022 17 November 2022	
Hospitalizations– no.^‡^	127,415	22,233	11,624	161,272
Cumulative hospitalisation rate per 100,000 population	822	143	75	1041
Proportion of total cumulative hospitalisations, %	79	13.8	7.2	100
Inferred infection: recorded hospitalisation ratio (95% CI)	65.9 (64.9–66.8)	457.3 (441.0–473.0)	Not applicable	
Period of recorded COVID-19 deaths, wave	31 March 2020 to 3 November 2021	4 November 2021 to 14 April 2022	15 April 2022 to 17 November 2022	
Recorded deaths in wave—no.	27,996	1802	913	30,711
cumulative recorded death rate per 100,000 population ^§^	180.6	11.6	5.9	191.7
Proportion of total cumulative recorded deaths, %	91.2	5.8	3	100
Inferred infection: recorded death ratio (95% CI)	299.7 (295.2–303.8)	5642.6 (5440.5–5836.3)	Not applicable	
Infection fatality risk (IFR) for recorded deaths (%)	0.33.	0.02	Not applicable	
Period of excess deaths wave	3 March 2020 to 27 November 2021	28 November 2021 to 19 March 2022	20 March 2022 to 17 November 2022	
Excess deaths in wave–no.	56,202	2974	6753	65,929
Cumulative excess death rate per 100,000 population	362.6	19.2	43.6	425
Proportion of total cumulative excess deaths, %	85.3	4.5	10.2	100
Inferred infection: excess death ratio (95% CI)	149.3 (147.1–151.3)	3719.0 (3296.5–3536.3)	Not applicable	
Infection fatality risk ^2^ (IFR) for excess deaths (%)	0.67	0.03	Not applicable	

^1^ The inferred number of infections in the population pre-Omicron BA.1 dominant wave was derived by multiplying the seroprevalence in unvaccinated individuals at the time of the pre-BA.1 serosurveys by the STATS-SA population [14]. The post-BA.1 inferred number of infections was obtained by multiplying the proportion of unvaccinated individuals showing overall serological evidence of SARS-CoV-2 infection (Appendix A) between the pre- BA.1 and post-BA.1-dominant wave serosurveys, by the STATS-SA population. ^2^ The infection fatality ratio was calculated as the inverse of the inferred infection to recorded deaths or excess ratios. All data are from the National Institute for Communicable Diseases daily databases [12] except for weekly excess deaths. Excess mortality from natural causes was defined per and sourced from the South African Medical Research Council [13]; the excess mortality data are reported through 4 June 2022. Other waves are lagged with respect to cases. Consequently, each of the hospitalization, recorded death, and excess death waves has its own cut-off points determining the start and end of the four epidemic waves. ^†^ Changes in testing rates, particularly the lower rates during Wave 1 due to constraints in laboratory capacity and prioritization of testing for hospitalized individuals, prevent direct comparisons, especially in terms of case numbers during the first wave in relation to the subsequent waves. Cases include asymptomatic and symptomatic individuals. Cumulative reported cases were sourced from the National Department of Health. ^‡^ Hospitalization data are from DATCOV, hosted by the National Institute for Communicable Disease, [12] as described previously [10,11]. The system was developed during the course of the first wave, with gradual onboarding of facilities; hence, these data could underestimate hospitalized cases in the first wave relative to subsequent waves. The hospitalized cases include individuals with COVID-19, as well as coincidental infections identified as part of routine testing for SARS-CoV-2 of individuals admitted to the facilities to assist in triaging of patients in the hospital. ^§^ Cumulative reported deaths were sourced from the National Department of Health.

## Data Availability

Requests for data sharing should be directed to Professor Shabir A. Madhi, email: shabir.madhi@wits.ac.za.

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
