# Peer review of "Sustained Low Incidence of Severe and Fatal COVID-19 Following Widespread Infection Induced Immunity after the Omicron (BA.1) Dominant in Gauteng, South Africa: An Observational Study"

_viruses, 2023, doi:10.3390/v15030597_

Round 1

Reviewer 2 Report

The authors have investigated a third cross-sectional population-based sero-survey in Gauteng, South Africa. The IgG seroprevalence of SARS-CoV-2 anti-nucleocapsid (anti-N) and anti-spike (anti-S) proteins was investigated.The authors have also evaluated the province for cases, hospitalizations, recorded deaths, and excess deaths. Excess deaths were evaluated from the inception of the pandemic through November 17, 2022. The paper is generally well written and structured, but in my opinion it has some shortcomings regarding some data analysis and text, My suggestions are as follows:

  1. The abstract should explain the entire manuscript, including its significance, attempts to solve a problem, methodology and approaches used in this study, summarised results, a conclusion, and implications. There are some points missing.
  2. The author should mention the ethical approval number in the manuscript.
  3. Based on the findings of the study, the authors may recommend an appropriate model.

Madhi, S. A., Kwatra, G., Myers, J. E., Jassat, W., Dhar, N., Mukendi, C. K., Nana, A. J., Blumberg, L., Welch, R., Ngorima-Mabhena, N., & Mutevedzi, P. C. (2022). Population Immunity and COVID-19 Severity with Omicron Variant in South Africa. The New England journal of medicine, 386(14), 1314–1326. https://doi.org/10.1056/NEJMoa2119658

Reviewer 3 Report

Summary

The authors report on the prevalence of SARS-CoV-2 spike and nucleocapsid antibodies across two cross sectional serological surveys conducted before and after the BA.1 wave of transmission in Guateng province, South Africa. They use these data to infer infection rates prior to and during the BA.1 wave which were then used to estimates infection fatality and excess mortality rates during the respective transmission waves. The key findings include (i) seroprevalence was very high in this population (~90%), largely driven by infection, at the post-BA.1 sampling timepoint and (ii) IFR and excess mortality declined substantially during transmission of predominantly BA.1 and Omicron sub-lineages versus pre-BA.1. The conclusion is high population-level immunity has markedly reduced the public health threat of SARS-CoV-2 in this setting. Although largely anticipated, the manuscript findings are important for characterizing population-level immune responses in South Africa, a population with low vaccine coverage, and provides a datapoint for better understanding global population immunity.  Overall the methods are reasonable, but clarification of several points is required, and improved presentation of the data would strengthen the paper.

Methods

1.     Please include in the main manuscript methods that serological levels were measured on Luminex (or MBA), and report the performance measures of the immunoassays.

2.     Please report the durability of antibodies measured with these assays (rates of seroreversion) if available. If not available, please detail this in the limitations.

3.     Please clarify which vaccines or vaccines were received (or if that is not available, at least what vaccines were available); and clarify if these vaccines stimulate immune response to both spike and nucleocapsid proteins or only spike.

Results

1.     Multiple sentences are confusing. For example L49-52 and L212. Please make a detailed review of the manuscript to improve readability.

2.     Include median days between surveys (plus median days between sampling for those with paired samples)

3.     There is a large volume of good data in Table 1, but the key data from this table should be summarized visually. For example, dot-whisker plots of pre- and post-BA.1 seroprevalence. And plots for “Overall serological evidence for SCVS infection” (rightmost column). Table 1 can then be moved to supplementary materials.

4.     Were any cases of seroreversion identified among participants with paired samples?

5.     Please include summary of the number of doses of vaccines received

Figures

1.     Please include shaded bars indicating the timing (and duration) of the respective surveys in figures 2 and 2.

2.     Figures 2 and 3 are hard to read. Removing every other month from the x-axis and rotating the months would help.

Round 2

Reviewer 2 Report

Manuscript can be accepted in present form.